# Home Office, Health Behavior and Workplace Health Promotion of Employees in the Telecommunications Sector during the Pandemic

**DOI:** 10.3390/ijerph191811424

**Published:** 2022-09-10

**Authors:** Zoltán Tánczos, Borbála Bernadett Zala, Zsolt Szakály, László Tóth, József Bognár

**Affiliations:** 1Department of Recreation, Hungarian University of Sports Science, H-1123 Budapest, Hungary; 2Doctoral School of Sports Sciences, Hungarian University of Sports Science, H-1123 Budapest, Hungary; 3Faculty of Health and Sports Sciences, Széchenyi István University, H-9026 Győr, Hungary; 4Department of Psychology and Sport Psychology, Hungarian University of Sports Science, H-1123 Budapest, Hungary; 5Institute of Sport Science, Eszterházy Károly Catholic University, H-3300 Eger, Hungary

**Keywords:** work health promotion, fitness, mental health and wellness in home office, multinational companies

## Abstract

Our study aims to present the perception and experiences of employees at a large multinational telecommunications company in Hungary working in home offices, as well as their health behavior and the workplace health promotion during the SARS-CoV-2 COVID-19 outbreak. The sample consisted of the full sample of highly skilled employees at a large telecommunication multinational company (N = 46). Throughout the analysis, tests for homogeneity of variance were followed by a MANOVA test to compare the groups’ means by gender, age, and job classification. The results clearly show that in the short term, workers’ mental health did not deteriorate, they do not argue or fight more with their partners and are no more depressed or irritable than before. Workers are less likely to think of ways to be more effective at work than in a home office. Similarly, they do not think that employers have more expectations than before the pandemic. Our research shows the assumption about home workers being less efficient or less diligent in their daily work to be false. A supportive and flexible employer approach to health-conscious employees will be an essential aspect in the future.

## 1. Introduction

The SARS-CoV-2 (COVID-19) pandemic has shaken the foundations of the global economy, bringing along serious individual, social and economic consequences [1]. In this situation, the labor market also had to react to the evolving environmental, societal, and personal challenges [2]. In times of contagious illnesses, home quarantine can be the first line of defense—for individuals and society alike [3]. 

It is well-known that quarantine has a rather negative impact on human relationships, health behavior and well-being and can cause psychological and mental issues such as irritability, insomnia, fear, and anxiety or concentration problems [4,5,6]. The COVID-19 pandemic changed daily routine, quality of life and way of work completely and globally [7]. According to restrictions, home quarantine had been enforced and their impact on adults’ health behavior, emotional and mental health, and well-being remain mainly undefined [8,9]. 

Even if the rate of home office workers has been typically high during the outbreak, there are significant differences among industrial sectors [10]. Experiences show that highly educated white-collar workers with high income were more likely to shift to working from home and maintain employment after the pandemic [11]. Working in home offices can generally benefit work efficiency; however, if the current situation persists for a longer period, it can unequivocally hinder private life, quality of life and relationships [12]. 

Shortly after COVID-19 restrictions, worsening health status, well-being and quality of life were observed, but focusing on healthy, active living on a daily basis had benefits for those working from home [13]. Employees switching to remote work believe that it will remain more common at their company even after the COVID-19 crisis [14]. In Hungary, the rate of remote work was 27.75% during the first wave of the pandemic [15].

### 1.1. Theoretical Background

The hedonic viewpoint on well-being, focusing on subjective well-being, is frequently equated with happiness and formally defined as having more positive effects and leading to greater life satisfaction [16]. The eudemonic viewpoint, in contrast, emphasizes psychological wellbeing, which is more broadly outlined in terms of a fully functioning person, and is operationalized either as a set of six dimensions [17], happiness plus meaningfulness [18], or as a set of wellness variables such as self-actualization and vitality [19].

Self-determination theory (STD) [19] is another perspective that has embraced the concept of eudemonia, or self-realization, as a core aspect of well-being. It attempted to specify what it meant to actualize the self and the way it can be best accomplished. Specifically, SDT suggests three basic psychological needs—autonomy, competence, and relatedness—and theorizes that the fulfilment of these needs is essential for psychological growth (e.g., intrinsic motivation), integrity (e.g., internalization and assimilation of cultural practices), and well-being (e.g., life satisfaction and psychological health), as well as experiences of vitality [20] and self-congruence [21].

SDT provides the concepts that guide the creation of policies, practices, and environments that promote both wellness and high-quality performance [22]. SDT most often considers seven aspirations that people may be pursuing as important over their lifetimes: financial wealth, recognition or fame, attractive image, personal development, meaningful relationships, community contributions, and physical fitness. Empirically, these aspirations revolve around two factors referred to as extrinsic aspirations and intrinsic aspirations. Research has also shown that when people place relatively strong importance on the extrinsic aspirations, and when they attain the extrinsic aspirations they desire, they tend to show signs of psychological ill-being, such as depression, anxiety, and low self-esteem, whereas when they pursue and attain intrinsic aspirations, they tend to show indications of psychological well-being, such as high self-actualization and self-esteem [23].

### 1.2. Health Behaviour and Workplace Health Promotion in Times of Pandemic

To respond to the pandemic, the World Health Organization (WHO) has proposed new guidelines on health and physical activity to avoid sedentary behavior and emphasize the importance of regular physical activity as a means to maintain lifelong health and overall well-being [24]. From the beginning of the millennium, changes in values, inactive lifestyle and the lack of health awareness have had a negative impact in terms of health and societal consequences for a large part of society [25].

Regular physical activity can be a useful means to preventing non-communicable diseases [26], and also can lower the level of depression, anxiety, and cognitive decline. Hence, the role of healthy active living is of high importance during pandemic situations [27]. Quarantine caused major interruption in quality of life and health behavior, which could result in increased intake of energy [28]. Altogether, these unhealthy behaviors negatively affect body weight, wellbeing, and also employee’s psychological and mental state. 

Increased office workload generally hinders employees from maintaining optimal health behavior, so employers play a crucial role in developing and promoting overall health [29]. Employees consider worksite wellness as a valuable employee benefit, but are dissatisfied with the wellness offerings their employers provide [30]. Supports offered by employers should target the needs and interests of the specific type of work and workforce [31].

To manage the risk of COVID-19′s spread, the fact that many employees were required to stay at home triggered teleworking practices. Telework proved to be the best solution to maintain the company’s operations while ensuring employees’ health and safety during the pandemic and securing an income for those in quarantine [3]. However, teleworking might result in employees working more because work–life boundaries are still blurred [32,33,34]. It can furthermore have a negative impact on employees’ mental and physical health [32,33,34], associated with a high risk of psychological distress and depression. Being away from both workplace and colleagues can make the employee feel isolated [35].

### 1.3. Purpose of Study

Businesses in the communication and telecommunication sector are more likely to support their employees’ health promotion with sports opportunities, improving fitness and health status or team-building activities. Employers believe that investing in occupational health promotion pays off in the long term as well-organized programs improve performance, boost morale and employee loyalty, which helps to retain a high-quality workforce in the long time [36].

Working from home can enhance flexibility; however, it comes with various challenges that have been substantially exacerbated during the COVID-19 pandemic. The level of these challenges is affected by gender, age, and job classification. As evident in health and work efficiency and affected by gender, age, and job classification [37]. It is known that sleeping disorders, work-related stress [38] and negative social practices has been proved as a consequence of teleworking [39]. Indeed, it can also certainly contribute to chronic diseases, such as diabetes, cardiovascular diseases, obesity, and hypertension [40]. However, as working from home for several months is unprecedented in our lives as employees, we can assume that this new situation might bring significant changes in work quality of life [41]. There is little research available on how employees in the telecommunication sector perceive their health status, quality of life quarantine and psychological well-being in Hungarian companies. The citizens’ need for telecommunication services generally strengthen during COVID and so the sector demonstrated different psychological, behavioral and work-related characteristics as compared to other sectors [42]. So, there is a need to further specify how the telecommunication sector copes with these challenges to behavior, psychological issues and working conditions. Consequently, the research question of the study is: what did the health-related characteristics of the employees of a telecommunication company demonstrate during the pandemic?

Hence, the purpose of this study was to demonstrate the perception and experience of employees at a large telecommunications company on health behavior and occupational health promotion while working from home during the SARS-CoV-2 COVID-19 pandemic. Major emphasis was placed on assessing the differences in attitudes to health, fitness and mental condition by gender, age, and job classification.

### 1.4. Hypotheses

It was assumed that employees of telecommunication sector working from home would demonstrate:mainly positive health status and behavior,a relatively positive emotional and mental state,a clear expectation in workplace health promotion;background variables (gender, age, and job classification) demonstrate meaningful differences in the results.

## 2. Materials and Methods

### 2.1. Sampling

Our sample was provided by a domestic affiliate of one of the largest international telecommunication companies in Hungary with 100 full-time employees, who regularly do office work but could also choose flexible work schedules even before the pandemic. The company provided a maximum of 4 days per month to the workers in the home office. Employees had the opportunity to work from home one day a week at their discretion. This opportunity was used by only 20% of the employees regularly, while 40% chose to work from home occasionally. Thus, long-term remote work, which was introduced during the pandemic, was unprecedented for the majority of the employees.

During the pandemic, all employees were working from home. The aim was to include every employee of the company but out of the total of 100, 46 workers answered all our questions voluntarily, and so were included in this study. The willingness of participation showed a similar tendency as in other international research, which remained close to 50% [43]. All of them had long-term contracts, were full-time workers and Hungarian citizens. The age range was between 20 and 57 years (M = 40.26 ± 10.52). For comparative analyses, the sample was divided by age (employees under and above 35 years), gender (19 female and 27 male) and job classification (38 employees or trainees, and 8 middle or top managers).

### 2.2. Measures

For the research, the cross-sectional research design was selected. This design enables us to analyze both the outcome and the exposures at the same time, and the small sample sizes have little effect on the results. The study was conducted as a pilot study for large-sample research. Data were collected through a questionnaire focusing on the level of agreement with attitude-related statements, using a 4-point Likert scale, where 1 denoted the lowest and 4 the highest level of agreement with the statements. We used the validated WHO Quality of Life (WHOQOL-BREF) questionnaire as a framework, which measures the quality of life. The questionnaire was modified to measure the workers’ quality of life in a quarantine situation and how the workplace can support its employees during the time of a pandemic. The data was collected during April 2020. The 36 questions encompassed home office (*n* = 23), workplace health promotion (*n* = 3) and health behavior (*n* = 10), with value 1 representing total disagreement and value 4 meaning total agreement. The questionnaire was filled voluntarily by 46 employees out of the total 100 workers of the examined unit.

### 2.3. Data Analysis

Data analysis was carried out with the IBM SPSS Statistics 22.0 software. Each item of the questionnaire was explored by descriptive statistics, presented in mean (M) and standard deviation (SD). For analyzing subscales, items were set in the identical direction, so that a higher score meant a healthier behavior. Responses of items in each subscale were summed. Effects of background factors (age groups, gender, and job classification) were analyzed using multivariate analysis of variance (MANOVA), regarding only the main effect. The significance level was set at *p* = 0.05, but tendencies were also considered at *p* < 0.1.

## 3. Results

### 3.1. Sample Characteristics

We aimed to target the particularities under quarantine (nutrition and stimulants under quarantine and work, emotional and mental effects, fitness, and health) (Table 1, Table 2, Table 3 and Table 4), general health behaviors (Table 5), workplace health promotion (Table 6), and differences by gender, age, and positions (Table 7, Table 8 and Table 9). The overall internal consistency of the 36-item questionnaire was acceptable (Cronbach’s Alpha = 0.623).

From the statements on nutrition and substance abuse (Table 1), most respondents ranked “In home office, I eat healthier than before” highly (M = 2.80 ± 0.91), in the 1st place. Also, coffee and alcohol consumption, as well as smoking, have not increased much during quarantine.

From the responses on home office during quarantine (Table 2), “Working in home office is suitable for me” (M = 3.13 ± 0.65) and “Quarantine is easy for me and is not a major problem” (M = 3.04 ± 0.76) were highly ranked. Employees do not think they would work more efficiently at the workplace than from home (M = 2.00 ± 0.87); similarly, they do not believe that expectations towards them would be higher than before the pandemic (M = 1.98 ± 0.80).

On the emotional impact side, it should be highlighted that employees are missing their life from before the pandemic (M = 2.41 ± 0.91). In addition, they are not bored, their mental health does not deteriorate, they do not argue or fight more with their partners, and they are not more depressed or irritable than before.

Among answers related to fitness and health in quarantine, the highest-ranked is “I can keep my life organized even in quarantine” (M = 3.07 ± 0.74). It can be stated that respondents do not suffer from joint pain, do not think they will gain weight, do not participate in less sport than before and their fitness is not deteriorating.

The everyday health behavior of workers is characterized by avoidance of harmful addictions (M = 3.35 ± 0.95), they generally feel well (M = 3.30 ± 0.70), take care of their health (M = 3.15 ± 0.67), are motivated to do sports (M = 3.11 ± 0.8) and enjoy their work in general (M = 3.02 ± 0.58). However, buying food based on price is ranked low (M = 2.09 ± 0.69).

Respondents think their workplace is open to health promotion (M = 3.26 ± 0.61). However, employee workload is at the medium level (M = 2.65 ± 0.77) and so is the motivation in participating in fitness assessments (M = 2.54 ± 0.91).

### 3.2. Differences by Background Factors

Descriptive statistics of the subscales are presented in Table 7.

As the main effect, MANOVA did not show any significant impact of gender and age; however, these sample groups seems to influence health behavior on a tendency level (Table 8).

A tendency was considered, and a univariate analysis of variance was conducted to reveal the subscales affected by job position (Table 9). Scores for the subscales “Nutrition and substance use during quarantine” and “Health behavior in everyday life” differed between the two job positions. Managers scored better in the former, while employees had higher scores in the latter subscale.

## 4. Discussion

Our study found that employees’ perception of the quarantine is less negative than in some previous research [44]. Respondents typically coped well with working under quarantine, did not typically experience increased difficulties in health behavior and their daily routines and lives did not deteriorate significantly. Studies proved that teleworking has positive aspects, for instance better work–life balance [45], quality of life [46] and also increased productivity [47].

The health problems associated with quarantine can be characterized by physical and psychosocial areas [48]. Earlier studies found that there were major negative changes in nutritional behavior and so body weight during COVID-19 [49]. It is known that sectors demonstrated differences on how they manage with health and work-related issues during the pandemic [50]. According to our findings, employees in this telecommunication company have not perceived major health problems and harmful addictions were typically rejected during the pandemic. The company’s employees are mostly forming a positive, motivated, and passionate community with members mostly free of addictions. This finding seems to underpin our first hypothesis stating that employees working from home demonstrate mainly positive health behavior in terms of both nutrition and substance use.

Consistent with self-determination theory, our results support the prevalence of autonomy and competence in our sample during the pandemic. Both personal health behavior and working conditions were perceived positively. Also, negative signs including ill-being, depression and anxiety were less apparent. Workers demonstrated aspirations mostly towards personal development and physical fitness.

It can be emphasized that there were no adverse changes in employees’ emotional and mental state because they seem to be satisfied, motivated and attentive to maintaining a health-conscious lifestyle. This is further reinforced by the fact that they considered themselves to be generally attentive to their well-being and health status. The results show that our second hypothesis about the relatively high level of emotional and mental state is also justified.

The employees clearly demonstrated trust in the company’s management in the field of workplace health promotion, which indicates competence and relatedness. It can also be of importance in this area that the occupational physician regularly checks the health status of the employees. However, most respondents indicated that they would not be happy to participate in a health behavior and fitness assessment. That probably means that employees feel autonomy in setting up and maintaining their health behavior program and do not need outside measures. This result supports the third hypothesis, which stated that employees working from home have a clear expectation of workplace health promotion programs.

It is important that the workers surveyed in this study generally feel well, take care of their health, and enjoy their work. Results show that typically positive health behaviors did not significantly decline during the epidemic. This proves that working from home can be as good an alternative as working in a traditional workplace. Men, young people, and employees tend to have more negative experiences and preferences of the epidemic, which is consistent with previous experience [8,14], as well as with our third hypothesis.

In our fourth hypothesis, it was stated that the background variables (gender, age, and job classification) would demonstrate meaningful differences. According to our results, gender and age did not produce significant differences in the results; however, job position had a slight difference, which might be explained by the unbalanced participation of managers and employees.

Although working from home can enhance flexibility, it comes with various social and psychological challenges such as cognitive overload and social isolation [32,33,34] that may negatively impact work productivity and well-being [51]. Thus, managers must find the balance between the work patterns. The sample of this study demonstrated self-realization to be a key aspect of well-being and quality of life. Autonomy, competence, and relatedness [19] were all apparent in the results with a high level of emotional and health-related stability, integrity, and well-being.

### Limitations

It is clear, however, that the workers’ health behavior was generally maintained and positive and did not significantly deteriorate during the epidemic. When assessing the results, it should not be overlooked that one telecommunication organization was included in this study, with the sample size being relatively low, and the standard deviations being typically high. Hence, our results cannot be generalized to different settings and a larger population; however, this research can open directions and points of view for further empirical studies.

## 5. Conclusions

From the perspective of the limitations and implications of this research, it would also be worth looking at the long-term risks of introducing home office in the future. Employees generally believe that telecommunications will play a more important role in the future as a secure source of income than traditional jobs [52]. Working from home used to be seen by employers and employees as a privilege that could be given as a reward, which before the quarantine, was available once a week at the examined telecommunications company [53]. Although, the changing working environment may lead to long-term health, mental, emotional, and social challenges.

Our research reveals that the assumption about home workers in this sector demonstrate mainly positive health status and behavior is acceptable. The workers also demonstrated an exceptionally positive emotional and mental state. However, it is suggested that monitoring of health behavior and the psychological and mental state is carried out during home office [54]. Our research also revealed that the employees had clear expectations regarding workplace health promotion. This result can be attributed to the preceding present work health promotion and recreational programs at the examined telecommunications company. It had been proved that a supportive work environment also helps to ensure employee’s performance and well-being [55]. The present research also identified further demands related to the work health promotion programs, such as nutrition, fitness and time management.

The background variables regarding to gender, age and job classification showed tendency level differences. Men, workers under 35 and employees seem to be more vulnerable to the drastic changes in the working environment. Based on our results, it is recommended that future work health promotion programs focus on these vulnerable groups. Larger scale and longitudinal studies could help us better understand the complexity and effects of different aspects and opportunities in employee’s health behavior and also worksite health promotion programs.

## Figures and Tables

**Table 1 ijerph-19-11424-t001:** Nutrition and substance use during quarantine (N = 46).

Statement	Ranking	Min	Max	Mean	SD
In home office, I eat healthier than before	1	1	4	2.80	0.91
In quarantine, I consume more coffee and/or stimulants than before	2	1	4	1.66	0.82
In quarantine, I consume more alcohol than before	3	1	3	1.54	0.64
In quarantine, I smoke more than before	4	1	2	1.17	0.38

**Table 2 ijerph-19-11424-t002:** Working in quarantine (N = 46).

Statement	Ranking	Min	Max	Mean	SD
Working in home office is suitable for me	1	2	4	3.13	0.65
Quarantine is easy for me and is not a major problem	2	1	4	3.04	0.76
I think quarantine is tolerable in the long term	3	1	4	2.67	1.03
More weekly overtime and longer working hours in quarantine	4	1	4	2.50	0.89
I think I work more efficiently at my workplace than in home office	5	1	4	2.00	0.87
More/greater expectations in quarantine than before	6	1	4	1.98	0.80

**Table 3 ijerph-19-11424-t003:** Emotional and mental characteristics in quarantine (N = 46).

Statement	Ranking	Min	Max	Mean	SD
I miss my old life	1	1	4	2.41	0.91
I feel duller and lazier in quarantine	2	1	4	1.72	0.81
I am more irritable and stressed in quarantine than before	3	1	4	1.65	0.82
I feel more depressed in quarantine than before	4	1	3	1.61	0.71
In quarantine, my partner and I argue more easily and more often, and we may even fight	5	1	3	1.57	0.65
In quarantine, I think my mental state is deteriorating	6	1	3	1.48	0.66
I often get bored in quarantine; I find it hard to tie myself down	7	1	3	1.48	0.69

**Table 4 ijerph-19-11424-t004:** Fitness and health in quarantine (N = 46).

Statement	Ranking	Min	Max	Mean	SD
I can keep my life organized even in quarantine	1	1	4	3.07	0.74
I spend more time sleeping in quarantine	2	1	4	2.20	1.07
The quarantine will certainly reduce my fitness level	3	1	4	1.93	0.98
I do less sport in quarantine than before	4	1	4	1.91	0.91
In quarantine, I will most probably gain weight, I am going to be fatter	5	1	4	1.67	0.84
My joints (spine, back, waist, hips, and other orthopedic complaints) hurt more in quarantine	6	1	4	1.59	0.75

**Table 5 ijerph-19-11424-t005:** Health behavior in everyday life (N = 46).

Statement	Ranking	Min	Max	Mean	SD
I usually avoid harmful addictions	1	1	4	3.35	0.95
I usually feel well	2	1	4	3.30	0.70
I usually take care of my health	3	1	4	3.15	0.67
I am usually motivated to do sports	4	1	4	3.11	0.88
I usually enjoy my work	5	1	4	3.02	0.58
I usually lead a physically active and sporty lifestyle	6	1	4	2.93	0.83
I usually pick groceries based on their ingredients	7	1	4	2.89	0.88
I usually have enough free time	8	1	4	2.61	0.88
There are sports facilities in my home (gym, garden, large terrace, sports court, swimming pool)	9	1	4	2.39	1.20
I usually pick groceries based on their price	10	1	3	2.09	0.69

**Table 6 ijerph-19-11424-t006:** Workplace health promotion (N = 46).

Statement	Ranking	Min	Max	Mean	SD
I think the management of my workplace is open to setting up and running an internal health promotion team	1	1	4	3.26	0.61
At my workplace workload management (e.g., deadlines, pace of work, workload) helps optimize the workload of employees	2	1	4	2.65	0.77
I would happily participate in a repeated, extended fitness assessment	3	1	4	2.54	0.91

**Table 7 ijerph-19-11424-t007:** Descriptive statistics of the subscales regarding background factors (mean ± SD).

	Sample (*n* = 46)	Gender		Age		Position	
Sub-Scale		Female (*n* = 19)	Male (*n* = 27)	<35 (*n* = 14)	35+ (*n* = 32)	Employee (*n* = 38)	Manager (*n* = 8)
Nutrition and substance use during quarantine	9.49 ± 3.44	9.60 ± 3.33	9.42 ± 3.57	8.17 ± 2.29	10.07 ± 3.73	8.97 ± 3.49	11.86 ± 2.04
Working in quarantine	17.39 ± 2.98	18.27 ± 2.37	16.83 ± 3.23	16.42 ± 2.97	17.82 ± 2.94	17.53 ± 2.94	16.71 ± 3.30
Emotional and mental impacts in quarantine	22.87 ± 3.71	23.80 ± 2.98	22.29 ± 4.05	21.75 ± 4.05	23.37 ± 3.51	22.97 ± 3.63	22.43 ± 4.32
Fitness and health in quarantine	18.05 ± 2.72	18 ± 2.95	18.08 ± 2.64	18.25 ± 2.38	17.96 ± 2.90	18.13 ± 2.74	17.71 ± 2.81
Health behavior in everyday life	29.56 ± 2.76	29.60 ± 2.67	29.54 ± 2.87	30.42 ± 2.97	29.19 ± 2.63	30.06 ± 2.63	27.29 ± 2.29
Workplace health promotion	8.64 ± 1.37	8.80 ± 1.08	8.54 ± 1.53	8.92 ± 1.51	8.52 ± 1.31	8.50 ± 1.30	9.29 ± 1.60

**Table 8 ijerph-19-11424-t008:** Results of the main effect of MANOVA.

MANOVA						
Factors	df	Approx. F	Trace Pillai	df1	df2	*p*
Gender	1	0.458	0.084	6	30	0.833
Age	1	0.994	0.166	6	30	0.447
Position	1	2.398	0.324	6	30	0.052
Residuals	35					

**Table 9 ijerph-19-11424-t009:** Results of analysis of variance.

ANOVA				
Sub-Scale	df1	df2	F	*p*
Nutrition and substance use during quarantine	1	37	4.412	0.043 ^1^
Working in quarantine	1	37	0.425	0.518
Emotional and mental impacts in quarantine	1	37	0.119	0.732
Fitness and health in quarantine	1	37	0.128	0.723
Health behavior in everyday life	1	37	6.680	0.014 ^1^
Workplace health promotion	1	37	1.946	0.171

^1^  *p* < 0.05.

## Data Availability

The data that support the findings of this study are available on request from the corresponding author, [ZT]. The data are not publicly available due to restrictions; they are containing information that could compromise the privacy of research participants.

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
