# Peer review of "Home Office, Health Behavior and Workplace Health Promotion of Employees in the Telecommunications Sector during the Pandemic"

_ijerph, 2022, doi:10.3390/ijerph191811424_

Round 1
Reviewer 1 Report
Given the high novelty and quality of this manuscript, dealing with a very important emerging issue with international resonance.
In particular, I do think that the paper is characterized by a high scientific soundness in the description of limitations, which do not allow any generalization but make this study a sort of pilot one that opens the way to future epidemiological studies on the same topic.
Here are a pair of ADDITIONAL COMMENTS for the AUTHORS: - I think that the Authors could add some comments about the nationality and the type of contract of the workers included in the study, highlithing possible outcome differences; - I do suggest to the Authors to consider the opportunity to add some comments about the role played by the occupational physician at the enterprise level in implementing workplace health promotion for home workers.
Author Response
Dear Reviewer,
First of all, we would like to thank you for your time and attention devoted to our research article. Your suggestions are very helpful and make the writing more consistent.
Your recommendations have been considered and our article has been corrected accordingly.
We have taken into account your comments and suggestions and made the following changes:
- We have completed the description of the limitations of the study, which states that it is a pilot study, which does not allow the generalization of the results.
- We added the requested details about the workers to the study.
We hope you will find our answers satisfactory.
Faithfully yours,
Authors
Reviewer 2 Report
Dear authors,
thank you for giving me the chance to read your manuscript "Home office, health behavior and workplace health promotion of employees in the telecommunications sector during the pandemic“.
The manuscript focuses to understand the factors for well-being of employees and managers in home office, presenting an actual topic.
I want to point out some aspects which I assume to need re-consideration to strengthen the manuscript:
Structure:
- The manuscript lacks structure & clarity. Individual parts & chapters are not logically connected, the text is not working as one entity.
Introduction:
- The introduction does not explain any research gap. There is no motivation for the research.
- There is no research objective.
- Study purpose is not logically evolved from the previous chapters.
- The hypothesis come out of nowhere. The review is very thin to allow for such hypothesis without critical emphasize. Please link the hypothesis to your text.
Research methodology:
- Which questions/ questionnaire was used?
- How did the authors come to the questions & how were these verified?
- Is the study sample valid at all? N=46 does not seem representative for one of the biggest IT-companies in Hungary.
Results:
- Table 1 to table 6: Are these information really needed in the main text?
- Table 7 & table 8: Please align your text & tables. Heading for table 7 stand alone.
- Table 8: Why is n=39? The abstract says n=46?
- Promised t-tests (abstract) are not present.
- MANOVA findings are not significant. They remain on a “tendency level“. What is the use of these findings?
Discussion:
- The findings of the authors should be discussed with regard to other study results in Hungary, Eastern Europe, Europe, globally. There are publications dealing directly with managers & employees, particularly in home office times.
Limitations:
- “… the workers’ private opinion …“ = self-biasedness?
- If the authors know, the results cannot be generalised: Why did they do this study? What is the purpose of it?
Conclusions:
- This chapter should be re-structured & compressed. It is somewhat twisted.
Formal:
- Please check the grammar & spelling once again.
I wish the authors all the best.
Best Regards
Author Response
Dear Reviewer,
First of all, we would like to thank you for your time and attention devoted to our research article. Your suggestions are very helpful and make the writing more consistent.
Your recommendations have been considered and our article has been corrected accordingly, especially considering methodological, conceptual and bibliographic aspects.
We have taken into account your comments and suggestions and made the following changes:
- We restructured the text as requested, and added paragraphs throughout the article to enhance its clarity.
- We specified the reseach gap, clarified the research question of the research as well as motivation of the study.
-We have reformulated the hypotheses as requested and added a wider literature review to the text.
- We specified which questionnaire was used for the study, the WHOQOL questioner was used extended with questions related to the pandemic. For the research topic, the above-mentioned WHO questionnaire was the most fitting, verified questionnaire.
- The study sample represents a company unit of 100 workers. The study concerns this unit, not the whole company, which is the biggest telecommunications company in Hungary. Clarification was added to the text.
- We aligned Table 7 and 8 as requested.
- The N sample changed due to a typo, which we corrected.
- We corrected the abstract, took out t-test analysis from it. The tendency level findings are important, because they show which examined group is more vulnerable to changes in the working environment, thus which groups needs more attention in work health promotion programs.
- We added the results of other related studies to the discussion. There is no similar analysis in regards to Hungary.
- We clarified the purpose of the study and its limitations.
- We shortened the Conclusions chapter and restructured it as suggested.
- We have the grammar and English checked.
We hope you will find our answers satisfactory.
Faithfully yours,
Authors
Reviewer 3 Report
The impact of the pandemic on workers' wellbeing will certainly remain an area of research interest for some time to come. In this particular paper, the researchers attempt to explore some aspects of this issue, but with the need for some revision in the following areas:
1. Introduction-it is not clear what exactly is the nature of the research problem, ie. whether it is wellbeing generally, health behavior or promotion. There is a need to better articulate this with some more substantive engagement of the literature.
2. Theoretical discussion on self-determination theory is thin and without a discussion of the relevance of the findings related to the area of interest, and the applicability/utility of this theory for the study.
3. The review of the literature in section 1.2 seems somewhat general with a discussion on several areas of research ie. physical activity, workload, quality of social relationship. What is the literature review trying to explain? What is the phenomenon being explored through the literature. This is not presented/clarified.
4. Focus/Industry selection-there is some focus here on the telecommunications industry. However, remote work is a critical aspect of this industry even before the COVID-19. What therefore is the rationale for selection within this study? This is not explained/presented. Further, the implications of this for the findings is not acknowledged, re no significant differences or distress.
5. methodology-there are some missing aspect of this. What is the research design? What are the scales used to inform the measures for the study? What are the reliability scores for these? What is the size of the industry? How was the sample selection? Why is the sample so small for a significant industry such as this? These questions must be addressed.
6. Discussion of the Data lacks substantive engagement of the literature. This can be significantly improved with some revision.
Author Response
Dear Reviewer,
First of all, we would like to thank you for your time and attention devoted to our research article. Your suggestions are very helpful and make the writing more consistent.
Your recommendations have been considered and our article has been corrected accordingly, especially considering methodological, conceptual and bibliographic aspects.
We have taken into account your comments and suggestions and made the following changes:
- We specified the reseach gap, clarified the research question of the research as well as motivation of the study in the introduction, and we also enhanced the literature review.
- We have improved the theoretical discussion on self-determination as requested.
- We clarified in the text, that the literature review explains phenomena examined by the used questionnaire.
- We specified the case selection process in the text. Remote work was not common in the case of this company before the pandemic.
- We extended the methodology as requested, and answered all the suggested questions in the respective sections of the article.
- We have expanded the literature review as requested.
We hope you will find our answers satisfactory.
Faithfully yours,
Authors
Round 2
Reviewer 2 Report
Dear authors,
thank you for giving me the chance to review your revised manuscript "Home office, health behavior and workplace health promotion of employees in the telecommunications sector during the pandemic“.
Even though I see the authors have done changes, I would still like to point out some aspects requiring strengthening:
Introduction:
- Are there other authors that have identified the research gap mentioned in the script?
- The hypothesis still do not have sufficient background. Hypothesis 2 says, the authors assume workers being sent home to have better mental health? Why – there are several studies showing the correlation of social distancing and mental health issues during home office? The authors should make clear within the introduction what they are after.
- The hypothesis are poorly formulated. Please check with other works.
Research methodology:
- The sample is not valid. 46 out of 100 does not meet the minimum sample quantity requirements.
Results:
- Table 6: Almost unreadible.
- Table 7: Does the table under table 6 belong there? I guess, it might have slipped.
- I still think, too many tables are hindering the flow of the text.
Discussion:
- This chapter still needs improvement. Even if there is no study from Hungaria available, the authors should discuss their findings in the light of similar studies.
- There are by far not enough references in this chapter.
Limitations:
- In scientific terms, I assume “… the workers’ private opinion …“ is expressed with the term self-biasedness?
- If the authors know, the results cannot be generalised: Why did they do this study? The study is not valid.
Conclusions:
- The conclusion misses the key findings, the contribution to the body of knowledge for theoreticians & practioneers, as well as the research gap to be targeted by future works.
Formal:
- The script is hard to read – the authors also printed the comments, not accepting changes from MS Word.
- The authors should add the added / changed text to their answer to the reviewer.
I wish the authors all the best.
Best Regards
Author Response
Dear Reviewer,
First of all, we would like to thank you for your time and attention devoted to our research article. Your suggestions are very helpful and make the writing more consistent.
Your recommendations have been considered and our article has been corrected accordingly, especially considering methodological, conceptual and bibliographic aspects.
We have taken into account your comments and suggestions and made the following changes:
Are there other authors that have identified the research gap mentioned in the script?
- We added a clafication requested ont he research gap in lines 128-136 as follows:
There is little research available on how employees in the telecommunication sector perceive their health status, quality of life quarantine and psychological well-being in Hungarian companies. The societies’ need for telecommunication services generally strengthen during COVID and so the sector demonstrated different psychological, behavioral and work related characteristics as compared to other sectors [42]. So, there is a need to further specify how telecommunication sector cope with these challenges on behavior, psychological issues and working conditions Consequently, the research question of the study is what health related characteristics of the employees of telecommunication company demonstrated during the pandemic.
- Alshaabani, A., Naz, F., Magda, R., Rudnák, I. Impact of Perceived Organizational Support on OCB in the Time of COVID-19 Pandemic in Hungary: Employee Engagement and Affective Commitment as Mediators. Sustainability 2021, 13(14), 7800. https://doi.org10.3390/su13147800/
The hypothesis still do not have sufficient background. Hypothesis 2 says, the authors assume workers being sent home to have better mental health?
- Hypothesis 2. reads as the following:
It was assumed that employees of telecommunication sector working from home demonstrate:
- a relatively positive emotional and mental state.
Why – there are several studies showing the correlation of social distancing and mental health issues during home office? The authors should make clear within the introduction what they are after.
- We have amended the introduction as advised. We clarified the research question in lines 134-136.
Consequently, the research question of the study is what health related characteristics of the employees of telecommunication company demonstrated during the pandemic.
The sample is not valid. 46 out of 100 does not meet the minimum sample quantity requirements.
- We addressed the comment as the following in the text:
The willingness of participation showed a similar tendency as in other international research, which remained close to 50% [43].
- Rongen, A., Robroek, S. J., van Ginkel, W., Lindeboom, D., Pet, M., and Burdorf, A. How needs and preferences of employees influence participation in health promotion programs: a six-month follow-up study. BMC Public Health 2014, 14(1), 1-8. https://doi.org/10.1186/1471-2458-14-1277
Table 6: Almost unreadible. Table 7: Does the table under table 6 belong there? I guess, it might have slipped.
- We realigned the tables. We assume that the corrections and added text causes the misalignment, because the template restricts the use of captions.
Discussion: This chapter still needs improvement. Even if there is no study from Hungaria available, the authors should discuss their findings in the light of similar studies. There are by far not enough references in this chapter.
- We modified the chapter as the following, with added references.
Our study found that employees’ perception of the quarantine is less negative than in some previous research [44]. Respondents typically coped well with working under quarantine, did not typically experience increased difficulties in health behavior and their daily routines and lives did not deteriorate significantly. Studies proved that teleworking has positive aspects, for instance better work-life balance [45], quality of life [46] and also increased productivity [47].
The health problems associated with quarantine can be characterized by physical and psychosocial areas [48]. Earlier study found that there were major negative changes in nutritional behavior and so body weight during COVID-19 [49]. It is known that sectors demonstrate differences on how they manage with health and work related issues during pandemic [50]. According to our findings, employees in this telecommunication company have not perceived major health problems and so harmful addictions were typically rejected during the pandemic. The company’s employees are mostly forming a positive, motivated, and passionate community with members mostly free of addictions. This finding seems to underpin our first hypothesis stating that employees working from home demonstrate mainly positive health behavior in terms of both nutrition and substance use.
Consistent with self-determination theory, our results support the prevalence of autonomy and competence in our sample during the pandemic. Both personal health behavior and working conditions was perceived positively. Also, negative signs including ill-being, depression and anxiety were less apparent. Workers demonstrated aspirations mostly towards personal development and physical fitness.
It can be emphasized that there were no adverse changes in employees’ emotional and mental state because they seem to be satisfied, motivated and attentive to maintain a health-conscious lifestyle. This is further reinforced by the fact that they considered themselves to be generally attentive to their well-being and health status. The results show that our second hypothesis about the relatively high level of emotional and mental state is also justified.
The employees clearly demonstrated trust in the company’s management in the field of workplace health promotion, which indicates competence and relatedness. It can also be of importance in this area that the occupational physician regularly checks the health status of the employees. However, most respondents indicated that they would not be happy to participate in a health behavior and fitness assessment. That probably means that employees feel autonomy in setting up and maintaining their health behavior program and do not need outside measures. This result supports the third hypothesis, which stated that employees working from home have a clear expectation of workplace health promotion programs.
It is important that the workers surveyed in this study generally feel well, take care of their health, and enjoy their work. Results show that typically positive health behaviors did not significantly decline during the epidemic. This proves that working from home can be as good an alternative as working in a traditional workplace. Men, young people, and employees tend to have more negative experiences and preferences of the epidemic, which is consistent with previous experience [8,14]. as well as with our third hypothesis.
In our fourth hypothesis, it was stated that the background variables (gender, age, and job classification) would demonstrate meaningful differences. According to our results, gender and age did not produce significant differences in the results, however, job position had a slight difference, which might be explained by the unbalanced participation of managers and employees.
Although working from home can enhance flexibility, it comes with various social and psychological challenges such as cognitive overload and social isolation [32,33,34] that may negatively impact work productivity and well-being [51] so managers must find the balance between the work patterns. The sample of this study demonstrated self-realization, as a key aspect of well-being and quality of life. Autonomy, competence, and relatedness [19] were all apparent in the results with a high level of emotional and health-related stability, integrity, and well-being.
- Ammar, A., Brach, M., Trabelsi, K., Chtourou, H., Boukhris, O., Masmoudi, L., Bouaziz, B., Bentlage, E., How, D., Ahmed, M., Müller, P., Müller, N., Aloui, A., Hammouda, O., Paineiras-Domingos, L. L., Braakman-Jansen, A., Wrede, C., Bastoni, S., Pernambuco, C. S., Mataruna, L., Hoekelmann, A. Effects of COVID-19 Home Confinement on Eating Behaviour and Physical Activity: Results of the ECLB-COVID19 International Online Survey. Nutrients 2020, 12(6), 1583. https://doi.org/10.3390/nu12061583
- Bartik, A., Cullen, Z., Glaeser, E., Luca, M., Stanton, Ch. What Jobs are Being Done at Home During the COVID-19 Crisis? Evidence from Firm-Level Surveys. SSRN Electronic Journal 2020, https://doi.org/10.3386/w27422
- Ryan, R. M., Deci, E. L. Intrinsic and extrinsic motivations: Classic definitions and new directions. Contemporary Educational Psychology 2000, 25(1), 54-67. https://doi.org/10.1006/ceps.1999.1020
- Gajendran, R. S., Harrison, D. A. The good, the bad, and the unknown about telecommuting: meta-analysis of psychological mediators and individual consequences. Journal of Applied Psychology 2007, 92(6), 1524. https://doi.org/10.1037/0021-9010.92.6.1524
- Henke, R. M., Benevent, R., Schulte, P., Rinehart, C., Crighton, K. A., & Corcoran, M. The effects of telecommuting intensity on employee health. AJHP 2016, 30(8), 604-612. https://doi.org/10.4278/ajhp.141027-QUAN-544
- Tavares, A. I. Telework and health effects review. International Journal of Healthcare 2017, 3(2), 30. https://doi.org/10.5430/ijh.v3n2p30
- Mann, S., Varey, R., Button, W. An exploration of the emotional impact of tele‐working via computer‐mediated communication. Journal of managerial Psychology. 2000, 15(7), 668-690. https://doi.org/10.1108/02683940010378054
- Practical Guide on Teleworking during the COVID-19 pandemic and beyond. A practical guide. Available online: https://www.ilo.org/wcmsp5/groups/public/---ed_protect/---protrav/---travail/documents/instructionalmaterial/wcms_751232.pdf (accessed on 25. August 2022).
- Montreuil, S. and Lippel, K. Telework and Occupational Health: A Quebec Empirical Study and Regulatory Implications. Safety Science, 2003, 41(4), 339-358. https://doi.org/10.1016/S0925-7535(02)00042-5
- Buomprisco, G., Ricci, S., Perri, R. and De Sio, S. Health and Telework: New Challenges after COVID-19 Pandemic. European Journal of Environment and Public Health, 2021, 5(2), em0073. https://doi.org/10.21601/ejeph/9705
- Al-Domi, H., Al-Dalaeen, A., Al-Rosan, S., Batarseh, N., & Nawaiseh, H. Healthy nutritional behavior during COVID-19 lockdown: A cross-sectional study. Clinical nutrition ESPEN, 2021, 42, 132–137. https://doi.org/10.1016/j.clnesp.2021.02.003
- Caligiuri, P., De Cieri, H., Minbaeva, D., Verbeke, A., Zimmermann, A. International HRM Insights for Navigating the COVID-19 Pandemic: Implications for Future Research and Practice. Journal of International Business Studies 2020, 51, 697-713. https://doi.org/10.1057/s41267-020-00335-9
- Lee, T.-C., Yao-Ping Peng, M., Wang, L., Hung, H.K., Jong, D. Factors Influencing Employees’ Subjective Wellbeing and Job Performance During the COVID-19 Global Pandemic: The Perspective of Social Cognitive Career Theory. Frontiers in Psychology 2021, 12. 577028. https://doi.org/10.3389/fpsyg.2021.577028
- Schmitt, J. B., Breuer, J., Wulf, T. From cognitive overload to digital detox: Psychological implications of telework during the COVID-19 pandemic. Computers in Human Behavior 2021, 124, 1-8. 106899. https://doi.org/10.1016/j.chb.2021.106899
Limitations: In scientific terms, I assume “… the workers’ private opinion …“ is expressed with the term self-biasedness?
- We rephrased the sentence as requested in lines 301-302.
It is clear, however, that the workers’ health behavior was generally maintained and positive and did not significantly deteriorate during the epidemic.
If the authors know, the results cannot be generalised: Why did they do this study? The study is not valid.
- We addressed the issue as follows in lines 305-307.
Hence, our results cannot be generalized to different settings and a larger population, however, it can open directions and points of view for further empirical studies.
Conclusions: The conclusion misses the key findings, the contribution to the body of knowledge for theoreticians & practioneers, as well as the research gap to be targeted by future works.
-We added the key findings in line 317-334.
Our research reveals that the assumption about home workers in this sector demonstrate mainly positive health status and behavior is acceptable. The workers also demonstrated an exceptionally positive emotional and mental state. However, it is suggested that monitoring of health behavior and the psychological and mental state is carried out during home office [54]. Our research also revealed that the employees had clear expectations regarding workplace health promotion. This result can be attributed to the precedingly present work health promotion and recreational programs at the examined telecommunications company. It is proved that a supportive work environment also helps to ensure employee’s performance and well-being [55]. The present research also identified further demands related to the work health promotion programs, such as nutrition, fitness and time management.
The background variables regarding to gender, age and job classification showed tendency level differences. Men, workers under 35 and employees seem to be more vulnerable to the drastic changes in the working environment. Based on our results, it is recommended that future work health promotion programs focus on these vulnerable groups. Larger scale and longitudinal studies could help us better understand the complexity and effects of different aspects and opportunities in employee’s health behavior and also worksite health promotion programs.
The script is hard to read – the authors also printed the comments, not accepting changes from MS Word.
- We unfortunately have not received the MS Word with the comments and changes. We would like to apologize for the inconvenience. With the correction we followed the instructions of the journal.
We hope you will find our answers satisfactory.
Faithfully yours,
Authors

Reviewer 3 Report
This paper is well improved. The authors must be commended. However, there are some authors that can be improved. These are as follows:
1. the introduction should state the gap being addressed in the paper. This can come towards the end of the last paragraph
2. The theoretical framework is improved and speaks to 7 aspirational measures. However, the authors need to show how this connects to the measures used in the questionnaire. The survey measures Quality of Life but does not show how it aligns to the theory. This can be improved.
3. The findings rank the preferences of the participants but the research questions speak of perceptions and not preferences. So this needs to be clarified.
Author Response
Dear Reviewer,
First of all, we would like to thank you for your time and attention devoted to our research article. Your suggestions are very helpful and make the writing more consistent.
Your recommendations have been considered and our article has been corrected accordingly, especially considering methodological, conceptual and bibliographic aspects.
We have taken into account your comments and suggestions and made the following changes:
- We have amended the introduction as advised. We clarified the research question in lines 134-136.
- We address how the questionnaire aligns to the theory in lines 177-179.
- We aligned the research question and the findings as suggested.
We hope you will find our answers satisfactory.
Faithfully yours,
Authors